# On the Hardness of Learning One Hidden Layer Neural Networks

**Shuchen Li**                                                    SHUCHEN.LI@YALE.EDU
**Ilias Zadik**                                                    ILIAS.ZADIK@YALE.EDU
**Manolis Zampetakis**                              EMMANOUIL.ZAMPETAKIS@YALE.EDU
*Yale University*

**Editors:** Gautam Kamath and Po-Ling Loh

## Abstract

In this work, we consider the problem of learning one hidden layer ReLU neural networks with inputs from $\mathbb{R}^d$. It is well known due to (Klivans and Sherstov, 2009) that without further assumptions on the distribution $\mathcal{D}$, e.g., when $\mathcal{D}$ can be supported over the Boolean hypercube, learning even one-hidden layer neural networks is impossible (or "hard"[1]) for polynomial-time estimators under standard cryptographic assumptions. Given the success of neural networks in practice, a long line of recent work has attempted to study instead the canonical continuous input distribution case where $\mathcal{D}$ is the isotropic Gaussian, i.e., $\mathcal{D} = \mathcal{N}(0, I_d)$ which is also the setting that we follow in this work. Yet, despite a long line of research, it remains open whether there is a polynomial-time algorithm for learning one hidden layer neural networks when $\mathcal{D} = \mathcal{N}(0, I_d)$. It is known that a single neuron, i.e., zero hidden layer neural network, can be learned in polynomial time (Zarifis et al., 2024), while neural networks with more than two hidden layers are hard to learn (Chen et al., 2022). Nevertheless, the case of one hidden layer neural networks is not well understood.

In this paper we close this gap in the literature by answering the question of efficient learnability of neural networks with one hidden layer. We establish that under the CLWE assumption from cryptography (Bruna et al., 2021), learning the class of one hidden layer neural network with polynomial size under standard Gaussian inputs and polynomially small Gaussian noise is indeed computationally hard. Importantly, solving CLWE in polynomial time implies a polynomial-time quantum algorithm that solves the *worst-case* gap shortest vector problem (GapSVP) within polynomial factors, a widely believed hard task in cryptography and algorithmic theory of lattices (Micciancio and Regev, 2009). En route, we prove the hardness of learning Lipschitz periodic functions under standard Gaussian inputs and polynomially small Gaussian noise. This improves the previous result from (Song et al., 2021), which proved the hardness for polynomially small *adversarial* noise.

We also utilize the more general reductions between CLWE and classical LWE due to (Gupte et al., 2022). In particular, we show that if we assume the hardness of GapSVP with subexponential approximation factors $2^{O(d^\delta)}$ for $\delta \in (0, 1)$, we can show the hardness of learning one hidden layer neural networks with polynomial size under Gaussian noise with $2^{-d^\eta}$ variance, where $\eta = \frac{\delta}{1+\delta} \in (0, 1/2)$. The current state-of-the-art algorithm for GapSVP is the celebrated Lenstra-Lenstra-Lovász (LLL) lattice basis reduction algorithm (Lenstra et al., 1982) which has approximation factor $2^{\Theta(d)}$. Hence, our results show that any polynomial time learning algorithm for one hidden layer neural networks for any variance of noise $\sigma^2 \geq 2^{-o(\sqrt{d})}$ would imply a major algorithmic breakthrough in the theory of lattices.

---

0. Extended abstract. Full version appears as (Li et al., 2024)

1. Following a standard convention, we refer to a computational task as "hard" if it is impossibile for polynomial-time methods.

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
