# OpenReview forum: "On the Hardness of Learning One Hidden Layer Neural Networks"
_algorithmiclearningtheory.org/ALT/2025/Conference — ALT 2025_

### Official Review · Reviewer_BbSU · 2024-11-07
**Solid theoretical paper, good fit for ALT, parts of presentation are convoluted, limited improvements and technical novelty**

**Rating:** 6
**Confidence:** 3

**Review:**

The authors proposed a reduction argument for computational hardness of weakly learning one-layer hidden neural networks via hardness of learning neurons and then the hardness of CLWE, which is a testing problem conjectured to be computationally hard. The major technical novelty comes from Theorem 8. The authors first constructed a CLWE instance by a slight Gaussian perturbation. Then the authors do a sample 2-split. One is used to train the learner; the other one is used to classify/validate. Now the author run the validation separately on the rest of the unused sample and some freshly harvest independent samples from the null. If the algorithm can weakly learn neurons and original samples are from CLWE’s alternative, then it seems the population risks should be lower, empirical risk concentrates well and the testing and reduction are done. Then the author constructed a neural net that can product the same the samples as the neurons and concluded the one-layer neural net’s learning hardness.

The scope of the topic is a good fit for ALT. In regards of writing, this is clearly a solid theoretical paper.

Some improvements or overhaul on the presentation can definitely be made by using more consistent notation among D,P,Q,h,A,\xi,z,y and their subscript and prime version. The notations are so convoluted given the technical complexity involved.  I will leave the confusions in writing to the end and move on to the technical part, some of which are probably also presentation related.

On the technical side:
I don’t understand the difference between \xi and \xi_0. I imagine \xi_0 is inherent inside phi and \xi is injected? If phi is Lipschitz and everything’s gaussian, injecting inside and outside should be similar? What’s the difference between F_xi and P_xi, only with no mod 1? Then how important is this mod 1? This is not explained clearly. An expert in CLWE setup might understand this but to an average person with no prior working knowletedge of it, this is far from clear. And where exactly in your algorithm did you use this injection and get P_phi? xi_i are what you injected? Or is P_phi just a proof artifact?

The novelty on the construction of the NN is not clear. This is perfectly fine by itself since it reduces the proof into proving the neuron case, but less so if that’s a defining differentiating feature from Song et al. Do you believe there’s something inherently hard to apply Song et al.’s approximation or it’s just you don’t know how. Have you tried other approximation results, or you don’t find any other works to be directly applicable? I did not see you mention any papers other than Song et al. If you have tried, maybe cite them and say they won’t work and why, which definitely boost the novelty! Right now, I tend to believe the novelty here on this construction is on the more incremental side. The neuron’s reduction is interesting, but it suffers presentation issues. Technically, this work is solid, not trivial but nothing ground breaking.

The following are some other confusions I had when reading this work.
Page 1: Long line of research with no citation?
Page 2: ``which arguably limits the generality of such an unconditional lower bound” Should cSQ be conditional lower bound as the authors suggested earlier? ``Specifically, in terms of conditional lower bounds… the so-called correlation Statistical Query (cSQ)...”
I suggest either add a few lines briefly explaining ``Cryptographic assumptions” somewhere early in the presentation, or refer the author to CLWE definition explicitly when refering to cryptographic assumtions. They were mentioned too many times, some are even with no explanation. ``cryptographic assumptions (specifically the learning with rounding assumption)”
Page 3: I suggest you modify ``the value of any such brittle algorithmic method in learning or statistics is unfortunately unclear”. Saying ``practically a non-negligible amount of noise always exists in these cases” or something like this should be enough to convey the idea, which I totally agree. I’m not affiliated with any of those authors nor offended. But what you wrote still seems unnecessarily rude and arrogant.
Fcal^NN_k has no equation number but is refered later on as (3). And in its definition, what is w_j and W?  Reader can guess via dimension but specifying it should be better.
Def of CLWE: Writing a sequence of decision problems {CLWE}_d is confusing (also the class notation). You are never considering a sequence of problems in this paper. You have one instance. Just say you are in the regime where d grows, and others depend on d may just be cleaner or am I missing something here?

**Paper Award:**

No

---

> ### Author Response · Authors · 2024-11-24
> **Thank you for the review! (Part 1)**
>
> We want to thank the reviewer for reading our paper carefully and for the constructive feedback and comments.
>
> Please see the General Comments post for a discussion on the technical contribution of our work. Please find our response to the additional comments that you raise below.
>
> > *“Do you believe there’s something inherently hard to apply Song et al.’s approximation or it’s just you don’t know how.“*
>
> We thank the reviewer for bringing up this point, we should have elaborated more on why it is in fact impossible to apply a standard approximation argument similar to the one used by Song et al. to transfer the hardness of the cosine neuron to hardness of NN under Gaussian noise. The reason is somewhat complicated, but we commit to adding an extended discussion in our revised version of our paper. We present also the key step where this approach fails below.
>
> First, recall that in their work Song et al. only prove results under “adversarial noise”. In particular, they start by establishing the CLWE hardness for learning the cosine neuron under any poly(d)-small adversarial noise and any sample size m=poly(d) [similar to our approach]. Then they consider any fixed sufficiently poly(d)-wide 1-hidden layer NN with poly(d)-small approximation error from cosine [notice that the approx error here only depends on the width of the NN, and classical results lead to approx error O(1/poly(width))]. Given that, any sample of the cosine neuron sample can be treated as an NN sample with poly(d)-small adversarial noise (the addition of the original adversarial noise and the newly added approximation error can be simply treated as poly(d)-small adversarial noise all together). This concludes their argument under adversarial noise: in particular, in this setting any efficient algorithm that weakly learns the NN with m=poly(d) samples weakly learns the cosine neuron with the same number of samples.
>
> In our case, achieving a similar guarantee under Gaussian noise is more tricky and the above simple idea of treating the approximation error as additional adversarial noise is naturally not fruitful. Moreover even trying to apply our Gaussianization idea fails for the following reasons. First notice again that, as we mention also above, the approximation error is some fixed poly(d)-small quantity, call it $\gamma$ that depends solely on the width of the NN used to approximate the cosine neuron [and not e.g., on the sample size used by any learning algorithm]. Given that, the above approximation idea applied to the cosine neuron leads to samples of the NN of interest with adversarial noise at least of order $\gamma$. Now trying to apply our main idea to “Gaussianize” the noise per new sample, we need to pay a price for a (slightly) larger noise variance per sample (but still poly(d)-small), call it $\sigma’$. Then following our Lemma 7 the total variation guarantee of this Gaussianization procedure for all the m=poly(d) samples together becomes (roughly) $m \cdot \gamma/\sigma’$, where m is the number of samples used by the algorithm is an arbitrary poly(d) quantity. In particular, as $\sigma=o(1)$, for the Gaussianization idea to work we at the very least need $m \cdot \gamma=o(1)$, something impossible to be guaranteed as m can be an arbitrary large poly(d) quantity and $\gamma$ is a fixed poly(d)-small quantity.
>
> It is finally worth mentioning that this led us to the second key step in our proof: to make our Gaussianization lemma to work we needed to find a NN with zero (or exp(d)-small) approximation error to a periodic and Lipschits neuron (something impossible for the cosine neuron). This led to our construction in Section 5.

---

> > ### Author Response · Authors · 2024-11-24
> > **(Part 2)**
> >
> > > *“On the technical side: I don’t understand the difference between \xi and \xi_0. I imagine \xi_0 is inherent inside phi and \xi is injected?”*
> >
> > Yes, in Lemma 7, $\xi_0$ is the noise from the CLWE sample, and $\xi$ is injected one for our reduction. We remind the reviewer that our goal is to show that $\phi(\gamma <w, x> + \xi_0) + \xi$ is indistinguishable from $\phi(\gamma <w, x>) + \xi$.
> >
> > > *“If phi is Lipschitz and everything’s gaussian, injecting inside and outside should be similar?”*
> >
> > No, injecting gaussian noise inside and outside are not similar. As a simple illustration of this, observe that injecting gaussian noise inside a bounded valued Lipschitz function, e.g., cosine, results in a bounded valued random variable. In contrast, adding gaussian noise outside results in a random variable with unbounded support.
> >
> > > *“What’s the difference between F_xi and P_xi, only with no mod 1?”*
> >
> > We don’t understand the confusion between $F_\xi$ and $P_\phi$ (we do not see any $P_\xi$ in our proof). $F_\xi$ is a randomized function and $P_\phi$ is a probability distribution.
> >
> > > *“Then how important is this mod 1? This is not explained clearly. An expert in CLWE setup might understand this but to an average person with no prior working knowletedge of it, this is far from clear.”*
> >
> > Mod 1 is very important for the hardness of CLWE, without mod 1 CLWE is just a linear regression problem.
> >
> > > *“And where exactly in your algorithm did you use this injection and get P_phi? xi_i are what you injected? Or is P_phi just a proof artifact?” *
> >
> > We are unsure what is the confusion of the reviewer on this, but we are happy to elaborate further in the revised version of our work if needed. In brief, our reduction from CLWE to weakly learning $\phi$ works by injecting the Gaussian noise $\xi_i$ after we apply $\phi$ to the label of the CLWE samples (see step 2 in the bottom of Page 8). We denote the resulting distribution by $P_1$ (if the samples come from the CLWE distribution) or $Q_1$ (if the samples come from the null distribution). Later our argument proceeds by proving that $P_1$ is close to $P_\phi$.
> >
> > > *“practically a non-negligible amount of noise always exists in these cases” or something like this should be enough to convey the idea”*
> >
> > We thank the reviewer for bringing up this comment. We are happy to follow the suggestion of the reviewer for rephrasing this part. We had no intention to offend any of the work in the exp-small noise regimes – we consider it a quite interesting and rich direction of mathematical work, that so happened to be not applicable to the point of view of our present work.

---

### Official Review · Reviewer_9Mcu · 2024-11-08
**PAC-learning of one-hidden-layer ReLU networks with Gaussian Label Noise**

**Rating:** 6
**Confidence:** 4

**Review:**

This paper considers PAC-learning of one-hidden-layer ReLU networks. For dimension $d\in\mathbb{N}$, consider $d$-dimensional Gaussian inputs $X\sim N(0,I_d)$ and labels $Y$ generated by a one hidden layer ReLU network of width $k = \widetilde{\omega}(\sqrt{d})$ with polynomially small Gaussian label noise. Then, the main result of this paper states that any $\epsilon$-PAC learner for this model, for $\epsilon = 1/poly(d)$, gives a poly-time quantum algorithm approximating GapSVP to within ${\rm poly}(d)$ factors, a conjecturally hard task. Notably, $k$ need not be too large (e.g., exponential) for the hardness, the result remains valid even when $k$ grows polynomially in $d$. The paper extends the results of Song et al. (2021) who established a similar hardness result for adversarially chosen noise to the (more natural) case of random noise. (Work by Song et al. studied periodic cosine neurons, but their results transfer to one-hidden-layer ReLU networks via standard approximation results).

Technical arguments appear very similar to Song et al. (2021) with the following additional technical steps: (a) injection of Gaussian noise to labels (Lemma 7), and (b) construction of an appropriate periodic neuron and a poly-width network agreeing on any bounded interval $[-R,R]$ (Lemma 10).

Strengths:

This is a nice result, yielding cryptographic hardness of PAC-learning one-hidden-layer ReLU networks with Gaussian label noise even when $k$ scales polynomially in $d$. Prior work showed hardness of learning two-hidden-layers and above, as well as the hardness of one-hidden-layers with adversarial noise, as opposed to random setting considered here. The paper is well-written and the relevant literature is adequately discussed.

Weaknesses:

The technical novelty over Song et al. (2021) feels somewhat marginal. The extension from adversarial to random noise appears to follow from nice yet relatively straightforward arguments: (a) injecting random noise (Lemma 7) and (b) considering a certain periodic neuron (Lemma 10).

Overall Recommendation:

I think the contribution of the paper is over the bar for ALT, so I recommend a weak accept. I’ll keep an open mind during rebuttals, and I encourage authors to elaborate especially on the technical novelty over Song et al. (2021).

Remarks/Questions:

— I think the hardness result against quantum algorithms should be highlighted further, particularly in the abstract.

— Display in Page 2: Please clarify that $(x)_+$ refers to ${\rm ReLU}(x) = \max\{x,0\}$.

— The authors have a discussion regarding noise level on Page 3. Can we say something about the width, i.e., what happens when $k=o(\sqrt{d})$? For instance, is it plausible to expect that there is a regime $d^c \ll k \ll \sqrt{d}$  in which one can show hardness, say by using an assumption other than CLWE?

Minor Typos:

— The notation for sphere $\mathcal{S}^{d-1}$ uses caligraphic S before display (2.3) and capital S at several other places (e.g. Lemma 7).

— Missing parenthesis on Page 7, $(x,\phi(\gamma\langle w,x\rangle+\xi)$.

**Paper Award:**

No

---

> ### Author Response · Authors · 2024-11-24
> **Thank you for the review!**
>
> We want to thank the reviewer for reading our paper carefully and for constructive feedback and comments.
>
> Please see the General Comments post for a discussion on the requested review on the width dependence of the state-of-the-art algorithms and our response for the technical novelty of the paper.
>
> We commit to fixing all the typos, we will emphasize the hardness against quantum computers, and we will clarify the notation $(x)_+$.

---

### Official Review · Reviewer_AUwi · 2024-11-08

**Rating:** 7
**Confidence:** 4

**Review:**

### Summary

This submission studies the question of learning 1-hidden-layer neural networks, with ReLU activation functions, under Gaussian inputs and noise. In particular, given i.i.d. samples of the form $(x, f(x) + z)$, where $x \sim N(0,I_d)$ and $z \sim N(0,1/\mathrm{poly}(d))$, the goal is to compute in polynomial-time *any* function $\hat{f}$ achieving squared loss better than the trivial mean estimator. They show that even achieving error that is better by an additive $1/\mathrm{poly}(d)$ (called weak learning) is impossible under standard cryptographic assumptions and when promised that the size of the network is polynomial (in particular, the width is at most polynomial in $d$).

### Strengths

This question has received significant attention from the learning theory community over the past few years, yet the question whether polynomial-sized networks can be weakly learned in the setting when the additive per-sample noise is Gaussian had remained open. Previous work implies a similar hardness result in the harsher setting in which the per-sample noise is similarly bounded in magnitude but otherwise adversarial (see footnote 2 in the submission).

The techniques used by the submission are simple but elegant. The hardness reduction starts from the continuous LWE (CLWE) problem and consists of two main steps: First, the submission observes that hardness of learning 1-periodic functions with adversarial (but bounded) per-sample noise follows immediately from CLWE. It then turns the noise into (close to, in total variation distance) Gaussian by simply adding Gaussian noise to each sample, since there are not too many samples, it is possible to take a union bound over all samples. Second, since this holds for any 1-periodic function, you can choose a piece-wise linear one and approximate this by a polynomial-sized ReLU network (by setting the function to 0 for large enough inputs).

The paper is very well-written and easy to follow.


### Weaknesses

The introduction could benefit from a more quantitative discussion of previous work, in particular, what is the dependence of the state-of-the-art algorithms on the width of the network and the weights? This can also come in a later place, e.g., after Theorem 1 and similar to the discussion on the noise rate.

### Typos

The exposition is generally polished, there are several minor typos.

Top of Page 2: that -> than
Second paragraph of Section 2.4: detecting feels like a strange choice of words here
Same place: The spacing after i.i.d. is off
Theorem 5: for solves -> for solving or that solves
After Eq. (4.1): Missing "to" after "we refer"
Subsequent paragraph: Missing parenthesis after "one can create a sample"
Several places: total variance distance -> total variation distance

**Paper Award:**

No

---

> ### Author Response · Authors · 2024-11-24
> **Thank you for the review!**
>
> We want to thank the reviewer for reading our paper carefully and for appreciating our results!
>
> Please see the General Comments post for a discussion on the requested review on the width dependence of the state-of-the-art algorithms.
>
> We commit to fixing all the typos.

---

### Author Response · Authors · 2024-11-24
**General Comments**

We first want to thank all reviewers for their helpful and encouraging feedback. We first address two questions shared by more than one reviewers and then proceed with responding to individual questions. We also corrected all the typographic mistakes found.

(A) Technical Novelty (raised by reviewers 9Mcu and BbSU)
------------------------------

It is correct that our argument to improve the result of Song et al. (2021) from adversarial noise to Gaussian noise is short and simple, something we are very open about in the paper. That said, we are kindly disagreeing with calling it “marginal”. Since 2021, multiple researchers have attempted this hardness question, something evident from the existence of (highly technical) works trying to handle the (easier case) of >=2 hidden layers (DV’21) and (Che+22) under no noise or (small) Gaussian noise. Furthermore, this hardness question (hardness of learning 1-hidden layer with Gaussian input and small Gaussian noise) has been informally but openly mentioned by experts as "the last missing piece in the computational complexity of learning NNs". It was to our surprise too that we found a simple and short way to conclude this important missing piece in the literature, as naturally for some time we were thinking about much more technical potential proof techniques. Yet, we honestly feel disappointed from hearing that a simple and short argument leading to an important result can be downgraded to a “somewhat marginal technical contribution” because of the complexity of the proof.


(B) Dependence on the Width of State-of-the-art Algorithms (raised by reviewers AUwi and 9Mcu)
---------------------------------------

We thank the reviewers for bringing up this topic. We agree that a more elaborate literature review on the topic would add value to the paper and we commit to including it in the revised version of our work.

The algorithmic guarantees (or positive results) on this work split into two main categories. There are (1) the tensor-decomposition based methods, e.g., [JSA15, GLM17, BJW19, GKLW19, ATV21], which (roughly) request the weights to be full-rank and the width k to be at most d and (2) dimension reduction approaches, e.g., [Dia+20] which (roughly) request that the weights are positive and that the width k is at most sqrt(log(d)).

We remind the reviewer that our hardness result applies under no assumption on the weights and whenever k>sqrt(d). In particular, our work implies as a simple consequence that the first class of tensor-based methods cannot be improved to lift the assumption of full-rankness at least when k>sqrt(d).

> *“For instance, is it plausible to expect that there is a regime $d^c \ll k \ll \sqrt{d}$ in which one can show hardness, say by using an assumption other than CLWE?”*

This is a good question. Unfortunately, the elaborate CLWE assumption does require us to assume k>sqrt(d) (which is though a particularly mild assumption given that no such hardness guarantee was known for any k=poly(d)). We are unsure how one could argue about the hardness when k=d^{a} for some a in (0,1/ 2) and we consider it a nice open question for future work.

---

### Meta-Review · Area_Chair_gq7K · 2024-12-14

**Recommendation:** Accept
**Confidence:** 5

**Metareview:**

There is a lucky uniformity of reviewers regarding the recommendations for the paper. All 3 reviewers recommend acceptance though at different level of enthusiasm. I am also happy to recommend acceptance.

**Paper Award:**

No